# Isolation and Full-Length Sequence Analysis of a Pestivirus from Aborted Lamb Fetuses in Italy

**DOI:** 10.3390/v11080744

**Published:** 2019-08-13

**Authors:** Enrica Sozzi, Antonio Lavazza, Alessandra Gaffuri, Fabio Carlo Bencetti, Alice Prosperi, Davide Lelli, Chiara Chiapponi, Ana Moreno

**Affiliations:** 1Istituto Zooprofilattico Sperimentale della Lombardia e dell’Emilia Romagna “Bruno Ubertini” (IZSLER), Via Antonio Bianchi 7/9, 25124 Brescia, Italy; 2Veterinary practitioner, Via Luigi Einaudi 35, 24049 Verdello (BG), Italy

**Keywords:** pestivirus, phylogenetic analysis, sheep, Italy, aborted fetus

## Abstract

Pestiviruses are distributed worldwide and are responsible for a variety of economically important diseases. They are not very host-specific, and thus sheep can be infected by well-known pestiviruses like bovine viral diarrhea virus (BVDV) and border disease virus (BDV), as well as by other recently discovered pestivirus species. The aim of this study is to describe the isolation and characterization of four pestivirus strains detected in aborted lamb fetuses from a single farm in the Brescia province (Northern Italy). A total of twelve aborted fetuses were collected and examined. After necropsy, organs were tested for the presence of infectious agents known as potential causes of abortion (*Brucella* spp., *Listeria* spp., *Coxiella burnetii*, *Chlamydophila* spp., *Mycoplasma* spp., *Neospora caninum*, and *Toxoplasma gondii*), and submitted to viral identification by isolation on Madin Darby bovine kidney (MDBK) cell culture and by PCR assay for Schmallenberg virus and pan-pestivirus RT-PCR real time assay. Three viral strains (Ovine/IT/1756/2017, Ovine/IT/338710-2/2017, and Ovine/IT/338710-3/2017) were isolated in the absence of cytopathic effects (CPEs) in cell cultures and identified with RT-PCR. Another pestivirus strain (Ovine/IT/16235-2/2018) was detected by PCR, but was not successfully isolated. Complete sequence genomic data of the three isolated viruses showed that they were highly similar, differed genetically from known pestivirus species, and were closely related to classical swine fever virus (CSFV). Beyond the identification of new ovine pestiviruses, this study indicates that a systematic diagnostic approach is important to identify the presence and map the distribution of both known and emerging pestiviruses.

## 1. Introduction

The genus *Pestivirus* within the family Flaviviridae has been recently classified, according to the International Committee on Taxonomy of Viruses (ICTV), into 11 viral species named from A to K, that include the four original species (bovine viral diarrhea virus (BVDV) type 1 (*Pestivirus A*), BVDV type 2 (*Pestivirus B*), classical swine fever virus (CSFV) (*Pestivirus C*), and border disease virus (BDV) (*Pestivirus D*)) as well as other pestiviruses recently classified as new pestivirus species [1].

Pestiviruses have positive-stranded RNA genomes with lengths of approximately 12.3 kb, which contain one large open reading frame flanked by 5′ and 3′ untranslated regions (UTRs). In the virus-encoded polyprotein, the viral proteins are arranged in the following order (from the N to the C terminus): Npro, C, E^rns^, E1, E2, p7, NS2-3 (NS2, NS3), NS4A, NS4B, BS5A, NS5B. The structural proteins are represented by the capsid protein C and three envelope proteins (E^rns^, E1, and E2). The remaining ones are non-structural (NS) proteins. 

Pestiviruses were first classified according to their host origin and the disease they cause. However, numerous investigations have proven that pestiviruses are not completely host-specific. It has been reported that BVDV can infect not only cattle but also sheep, swine, goats, and numerous other ungulate species [2,3]. In addition, it has been demonstrated that BDV can infect sheep, swine, and goats [4,5]. Only CSFV appears to be restricted to a single host species, except in experimental studies, in which CSFV can infect and may cause disease in cattle and goats [6]. 

Pestiviruses are distributed worldwide and are responsible for a variety of economically important diseases. Moreover, in the last two decades, a growing number of new pestiviruses have been discovered in different domestic and wild animal species. Pestivirus infections may be subclinical or produce a range of clinical conditions characterized by acute diarrhea, acute hemorrhagic syndrome, acute fatal disease, and wasting disease. Moreover, transplacental infection, which can result in fetal death, congenital abnormalities, or lifelong persistent infection, may also play an important role in the epidemiology of pestiviruses in swine and ruminants.

This study describes the isolation and complete genome sequence of three pestiviruses detected in aborted lamb fetuses from a single farm located in the Brescia province of Northern Italy. A further pestivirus was only detected by PCR but not successfully isolated in cell culture. Sequence data of these viruses showed that they cluster together and separately from any other established and proposed tentative ovine pestiviruses.

## 2. Materials and Methods

### 2.1. Clinical Cases and Sampling

Between January 2017 and February 2018, cases of abortion in the last trimester of pregnancy were reported in sheep from a farm located in the Brescia province of Northern Italy. In that farm, approximately 2000 lactating Assaf breed sheep were present, and continuous introductions of animals from Spain were reported. A total of twelve aborted fetuses were collected and submitted to the Istituto Zooprofilattico Sperimentale della Lombardia e dell’Emilia Romagna (IZSLER). The first two fetuses were twins from a single abortion and were submitted in January 2017; thus, the samples were treated as a single submission, i.e., the organs taken from both fetuses were pooled prior to homogenization. The other ten fetuses were submitted between December 2017 and February 2018. During this period, about 5%–7% of the pregnant sheep experienced an abortion. At necropsy, the samples of lungs, spleen, liver, and kidneys from each fetus were collected and pooled together. 

### 2.2. Laboratory Investigation 

Post-mortem examinations were conducted on all 12 aborted fetuses. For virological investigations, organs were homogenized and prepared in minimum essential medium (MEM; Gibco, Life Technologies, Paisley, UK) supplemented with antibiotics (1000 U/mL penicillin, 1 mg/mL streptomycin; Gibco, Life Technologies, Paisley, UK) and anti-mycotic agents (2.5 µg/mL amphotericin B; Gibco, Life Technologies, Paisley, UK). 

In order to detect pestivirus, viral RNA was extracted from 200 µL of homogenized pools of organs by Trizol. Pan-pestivirus real-time RT-PCR was performed according to Office International Epizooties protocol [7]. Then, the same pools of organs’ homogenates were inoculated in confluent monolayers of bovine primary kidney and Madin Darby bovine kidney (MDBK) cells, previously tested as free from contamination with any endogenous pestivirus. Cell culture medium (MEM; Gibco, Life Technologies) was supplemented with gamma-irradiated fetal bovine serum, free of pestivirus antibodies, antigens, and genomes. The infected cells were incubated at 37 °C with 5% CO_2_ and checked daily for cytopathic effects (CPEs). Two “blind” passages were performed, each after 6–7 days of incubation. Then, after one week of incubation, the cell lysate from the third passage was harvested and tested by RT-PCR real-time assay to assess the isolation of pestivirus. 

All the organ samples collected from aborted fetuses were also tested for the presence of other infectious agents known as potential causes of abortion. For bacteriological agents, such as *Brucella* spp., *Listeria* spp., and *Mycoplasma* spp., the screening and pathogen identification were conducted according to OIE standardized protocols [8]. For *Coxiella burnetii*, the method described by Berri et al. [9] was used. The presence of *Chlamydophila* spp. was investigated by real-time PCR directly in biological samples [10] and typing by PCR-RFLP assay, targeting the 16S ribosomal gene [11]. For *Mycoplasma* spp., the PCR method described by van Kuppeveld et al. [12] was used. Furthermore, PCR tests to detect *Neospora caninum* and *Toxoplasma gondii* were conducted [13,14]. Finally, direct molecular detection through PCR assay was conducted for Schmallenberg virus (SBV) as described by Schulz et al. [15].

### 2.3. Molecular Analyses

For sequencing, all the positive samples in the pan-pestivirus RT-PCR and the cell culture viral isolates were further amplified using primers described by Letellier et al. [16] to obtain the 5′-UTR fragments. Alignment and phylogenetic analyses were conducted on the 5′-UTR fragments (287 bp).

The complete genomes were obtained from both the homogenized organs (EO) and the cell culture isolates (CC) using the MiSeq platform (Illumina, San Diego, CA, USA). Sequencing libraries were made with an Illumina TruSeq RNA Library Preparation Kit v 2 according to the manufacturer’s instructions. Reads of samples Ovine/IT/1756/2017 and Ovine/IT/338710-2/2017 were de-novo assembled by CLC Genomic Workbench v.11 (QIAGEN, Milan, Italy), with an average coverage of 26 and 40, respectively. The full-length genome of sample Ovine/IT/338710-3/2017 was obtained by mapping reads against Ovine/IT/1756/2017 as the reference (average coverage was 6).

Nucleotide sequences were aligned and compared with sequences available in GenBank (www.ncbi.nlm.nih.gov) using Lasergene software v 10.0 (DNAStar, Madison, USA). Nucleotide BLAST (blastn) and protein BLAST (blastp) algorithms (http://blast.ncbi.nlm.nih.gov/Blast.cgi) were employed to find per cent identities. 

Phylogenetic analysis was performed using the MEGA6 and IQ-tree softwares [17,18], and complete genome sequences of viral strains were compared with other pestivirus sequences belonging to BDV, BVDV-1 and -2, CSFV, and unclassified pestiviruses originating from Europe, America, and Asia obtained from the NIAID Virus Pathogen Database and Analysis Resource (ViPR) [19] online (http://www.viprbrc.org). Complete genome pestivirus sequences and 5′-UTR sequences were aligned with the corresponding reference sequences by MUSCLE from theViPR [19]. A maximum likelihood phylogenetic tree was constructed using the IQ-tree software by applying the HKY85 model of nucleotide substitution, identified using ModelFinder selection [20]. Similarity plots of pestivirus complete genomes were generated by SSE v1.2 using a sliding window of 600 and a step size of 100 nucleotides.

Putative recombination events were verified using the Recombination Detection Program 4 (RDP4) software (http://web.cbio.uct.ac.za/~darren/rdp.html), using five different methods. A recombination event was accepted only when detected by ≥3 methods implemented in the program with a *p*-value <0.05. 

## 3. Results

### 3.1. Clinical Cases 

During post-mortem examinations, three fetuses were found mummified, while nine were well formed. Out of these nine, we did not detect any lesions in only two fetuses. In the remaining seven fetuses, we observed, sometimes in association, the following lesions: Hairy fleece with abnormal yellow pigmentation, live degeneration and systemic lesions characterized by hemorrhagic subcutaneous edema, and the presence of serum hemorrhagic fluid in pleural and peritoneal cavities. The details of the lesions found in each fetus necropsied are reported in Table 1.

### 3.2. Virus Isolation and Identification

On the whole, during the study, three viral strains (Ovine/IT/1756/2017, Ovine/IT/338710-2/2017, and Ovine/IT/338710-3/2017) were isolated in the absence of CPEs in cell cultures. All viral isolates, in fact, were identified with RT-PCR by the amplification of a specific product of 287 bp with primers BE and B2 [16]. Another fetus appeared to be pan-pestivirus positive (Ovine/IT/16235-2/2018), but we did not succeed in isolating the viral strain. Out of the 12 tested lamb fetuses, six, including the Ovine/IT/338710-2/2017 and Ovine/IT/338710-3/2017 fetuses, were also positive for *Chlamydophila abortus* (Table 1).

### 3.3. Genome Characterization

PCR sequences for the 5′-UTR fragments were obtained for Ovine/IT/1756/2017, Ovine/IT/338710-2/2017, Ovine/IT/338710-3/2017, andOvine/IT/16235-2/2018, while the complete genome was obtained for both the EO and the CC of Ovine/IT/1756/2017 and for the EO of both Ovine/IT/338710-2/2017 and Ovine/IT/338710-3/2017. The three full genome sequences from the Eos were deposited in GenBank (NCBI) with the accession numbers MG770617, MK618725, and MK618726.

Phylogenetic analysis based on partial sequencing of 5′-UTR fragments identified six different clades, corresponding to CSFV, BDV, BVDV, Aydin PeV, Tunisian pestiviruses, and atypical porcine pestivirus. In particular, the ovine viruses detected and analyzed in this study formed a different viral group closely related to the classical swine fever (CSF) isolates (Figure 1).

The phylogenetic relationships of the complete genomes produced a phylogenetic tree with six highly supported monophyletic clades, corresponding to CSFV, BDV, BVDV-1, BVDV-2, BVDV-3, and giraffe virus, and four divergent lineages, corresponding to Pronghorn virus, rat pestivirus, Bungowannah virus, and porcine atypical pestivirus.

No correlation was observed between pestiviruses isolated from lamb fetuses in this study and strains previously isolated from small ruminants in Italy, which were classified as a subtype of the BDV species [21] and into Tunisian-like pestiviruses [22]. Again, these ovine Italian sequences formed a different viral group inside the *Pestivirus* genus that clustered together with the classical swine fever (CSF) isolates (Figure 2).

The percentage of similarity of pairwise distances confirmed the results obtained by phylogenetic analysis (Table 2). All nucleotide sequences of the new Italian strains identified in this study formed a distinct cluster that was closely related to the CSF species (72.2%–72.3%and 87.0%–91.2%for the complete genome encoding the polyprotein and the 5′-UTR, respectively). Compared with other pestivirus species, a percentage of identity <70% was identified for the complete genome encoding the polyprotein, but when taking into account the 5′-UTR, a higher percentage of identity was observed with the small ruminant pestiviruses, including BDV, and Tunisian and Turkish pestiviruses (75.8%–87% of identity). Interestingly, the new Italian strains showed the highest identity percentage value (94.8%–98.6% for the 5′-UTR region) with the Spanish isolate 5440/99 (GenBank accession no. AY159514), whose homology with CSFV reference strain AlfortA19 (GenBank accession no. U90951) and Brescia (GenBank accession no. AF091661) was 90.7%. This strain was identified in a stillborn lamb from a farm in Spain, which was characterized by a high incidence of abortions, mortality in lambs, birth of weak lambs, and immunosuppression [23]. 

Thereafter, the nucleotide sequence of Ovine/IT/1756/2017 was compared to the complete genome nucleotide sequences of CSFV (strain Alfort/187, GenBank accession no. X87939), as well as related Aydin-like pestiviruses (strain Aydin/04, GenBank accession no. JX428945 and Burdur/05, GenBank accession no. KM408491). The sequence of the “new” Italian pestivirus showed a higher sequence identity to CSFV across the whole genome than to the Aydin-like pestivirus sequences (Figure 3).

Recombination analysis did not detect any recombination event in the Italian Ovine/IT/1756/2017 pestivirus. 

## 4. Discussion

In this study, three highly similar pestiviruses were isolated at different times points from four aborted lamb fetuses originating from a single farm in Northern Italy. A fourth positive case was detected by using the sole 5′-UTR PCR, but it was not possible to obtain the whole genomic sequence, nor to isolate it in vitro.

Co-infection with *Chlamydophila abortus* was present in two out of the four pestivirus-infected fetuses, so the direct responsibility of the pestivirus in abortion could not be definitively established. However, the role of the pestivirus as an immunosuppressive agent, exacerbating the pathogenicity of co-infecting microorganisms, should also be considered according to previous studies [24].

Traditionally, all pestiviruses isolated from goats and sheep have been referred to as BDV, but genetic characterization has shown that these species can be infected with a wide range of pestiviruses including BDV, BVDV-1, and BVDV-2 [21]. Moreover, recent genetic and antigenic analyses determined that ovine and caprine pestiviruses were not monotypic, but belong to two separate species, the Aydin-like pestiviruses (*Pestivirus* I) and the newly proposed species, Tunisian sheep pestiviruses.

The sequencing analysis of the 5′-UTR and the complete genome permitted us to definitively assign the identified strains in this study to a group within the *Pestivirus* genus, closely related to the CSFV isolates, but it was evident as well that they formed a distinct cluster that originated from a branch common to CSFV isolates. Remarkably, analysis of the complete sequence showed that these pestiviruses formed a well-supported single clade, demonstrating their genetic difference from the established and proposed pestivirus species of sheep. In fact, they did not cluster with any of the BDV subtypes, Aydin-like pestivirus, or Tunisian sheep pestiviruses. Moreover, the occurrence of a similar pestivirus was reported in 1999 in Northern Spain, where a CSFV-like viral strain was detected in ovine samples taken from a flock suffering from abortions and births of stillborn and weak lambs with nervous symptoms. In that case, lesions associated with a fungal infection were also shown in the lungs of a stillborn lamb [23]. Furthermore, the phylogenetic analysis of the 5′-UTR, i.e., 94.8%–98.6% nucleotide identity, confirmed the quite close relationship between this Spanish strain and the Italian ovine pestiviruses. Such a finding of a similar strain in Spain, even considering the long interval of time that has elapsed between the two detections, might induce consideration of the trade of sheep among the two countries as a possible source of the infection, and emphasizes once more the importance of health control of imported and exported animals. Therefore, the fact that the Spanish strain appears to group similarly to the Italian strains when analyzed phylogenetically would suggest that it must be grouped with them for taxonomic purposes. 

What is certainly unclear is the origin of these strains, since no CSFV strains other than the Spanish isolate have been previously detected in sheep or in any other species. Thus, even if it can be suggested that sheep could have been a spillover host from pigs, especially when CSF was endemic in domestic swine and wild boars, the opposite, i.e., a species jumps from ovine to pigs, cannot be excluded. In fact, in the previous study of Rios et al. [25], it was suggested that CSFV likely emerged 225 years ago when the Tunisian sheep pestivirus jumped from its natural host to swine. This study reinforces the hypothesis that unknown pestiviruses, more or less related to CSFV, could exist, and need to be identified and better characterized. In addition, their role as a potential source of new virulent agents, such as through species jump and/or genetic changes (including recombination, which was not detected in this specific case), cannot be excluded and should be further investigated. 

Whether these Italian ovine pestiviruses, as well as the Spanish strain, should be considered a new species is not completely defined. In fact, the International Committee on Taxonomy of Viruses (ICTV) (https://talk.ictvonline.org/) considers different parameters to define species demarcation for pestiviruses, e.g., nucleotide sequence relatedness is an important criterion, and in most cases, a degree of genetic variation over 15% within the 5′-UTR or at least 25% for the complete genome allows species demarcation. However, when the nucleotide sequence relatedness is ambiguous, it should be complemented with additional comparative data, such as cross-neutralization tests (at least a 10-fold difference in neutralization titre), host of origin, and type of disease. Therefore, although the Italian sheep pestiviruses showed levels of divergence near the cutoff levels (8.8%–13% in the 5′-UTR and 27.8% across the whole genome), but which were not clearly over the threshold suggested by ICTV to warrant the classification of “new” species, the differences in host of origin and type of disease/lesions detected could meet the criteria for the definition of a novel pestivirus. 

Nevertheless, other criteria may support recognition as a novel species and should be considered for a better and more definitive pestivirus species characterization. One may be the full description of clinical signs and pathological lesions, which our preliminary observations indicate to be quite different in sheep from those observed during CSF outbreaks in pigs, likely through experimental reproduction. Other aspects, which are actually included in our ongoing and future studies, include the definition of *in vitro* properties, such as the presence of CPEs and the requirements of different cell lines, including those of pig origin, and the study of antigenic relatedness, including cross-neutralization tests.

More generally, beyond the identification of new ovine pestiviruses, it could be certainly stated that the systematic diagnostic approach adopted in this study is important to identify the presence and map the distribution of both known and emerging pestiviruses in Italy from the different domestic and wild susceptible species. Indeed, this could help to define the epidemiological patterns and the evolution of pestivirus infections and relative diseases. In fact, more information is needed in order to clarify the correlation of genetic and antigenic changes with virulence and clinical signs, and with epidemiological factors, such as origin, host range, and the geographic distribution of cases.

## Figures and Tables

**Figure 1 viruses-11-00744-f001:**
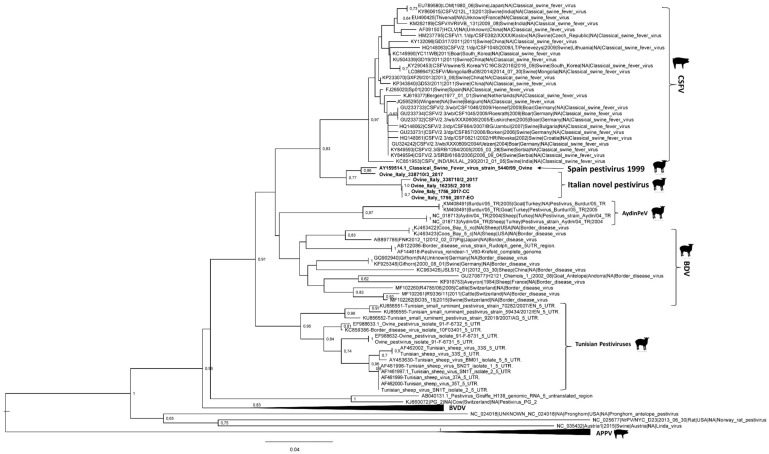
Maximum Likelihood phylogenetic tree constructed with five 5′-UTR sequences of the four pestiviruses identified in this study (Ovine/IT/1756/2017-EO, Ovine/IT/1756/2017-CC, Ovine/IT/338710-2/2017, Ovine/IT/338710-3/2017, and Ovine/IT/16235-2/2018) and 100 pestivirus sequences retrieved from Genbank. Bootstrap values above 60% (1000 replicates) are shown. The novel Italian pestivirus strains and the Spanish strain isolated in 1999 are reported in bold.

**Figure 2 viruses-11-00744-f002:**
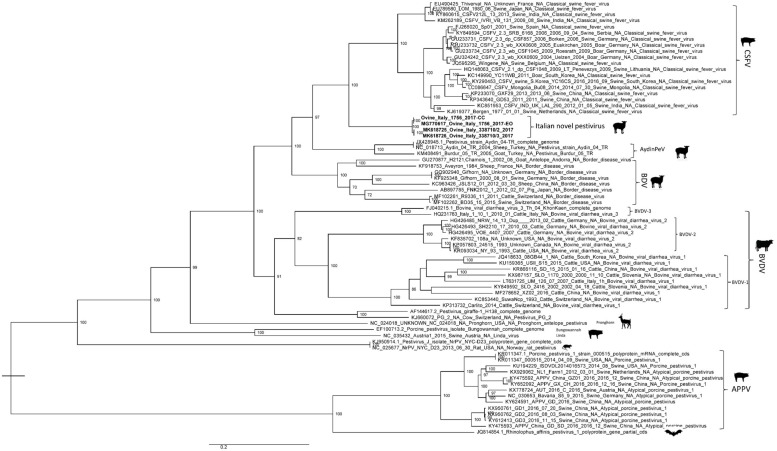
Maximum Likelihood phylogenetic tree, based on the complete genomes of both the EO and CC sequences of Ovine/IT/1756/2017, Ovine/IT/338710-2/2017, and Ovine/IT/338710-3/2017 (in bold) and 68 other pestivirus sequences. The best-fit model according to BIC was HKY85. The data of pestivirus isolates shown in the phylogenetic tree consisted of GenBank Accession No., viral name, country, and date of isolation.

**Figure 3 viruses-11-00744-f003:**
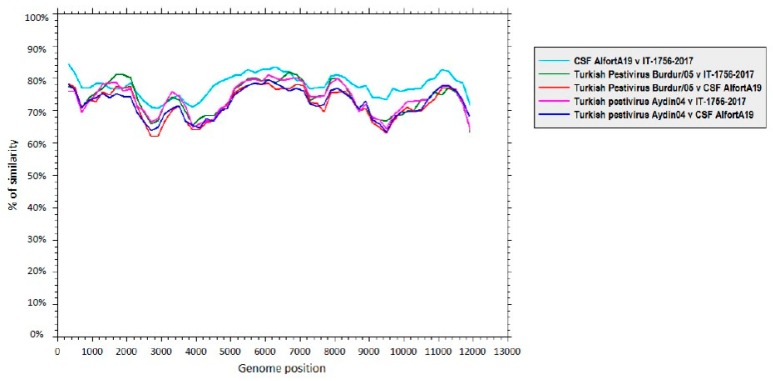
Sequence identity between the complete genome of Ovine/IT/1756/2017 and the closely related pestiviruses, i.e., classical swine fever virus(CSFV; strain Alfort/187, GenBank accession no. X87939) and Aydin-like pestiviruses (strain Aydin/04,GenBank accession no. JX428945 and Burdur/05, GenBank accession no. KM408491). Similarity plots were generated using SSE version 1.2, using a sliding window of 600 and a step size of 200 nucleotides.

**Table 1 viruses-11-00744-t001:** Lamb fetuses examined in this study, lesions detected, and results for pestivirus on both original extracts (EO) and cell culture (CC), and *Chlamydophila abortus*.

Sample Id.	Lesions	Laboratory Outcome		
		Pestivirus		*Chlamydophila abortus*
		EO	CC	
1756/2017 (1+2) pool	(1) mummified (2) hairy fleece with abnormal yellow pigmentation	+	+	negative
338710-1/2017	hemorrhagic subcutaneous edema, serum hemorrhagic fluid in cavities	−	−	positive
338710-2/2017		+	+	positive
338710-3/2017		+	+	positive
350462/2017	no lesions	−	−	negative
350530/2017	no lesions	−	−	negative
16235-1/2018	hemorrhagic subcutaneous edema, serum hemorrhagic fluid in cavities	−	−	negative
16235-2/2018	hairy fleece with abnormal yellow pigmentation	+	−	negative
30564-1/2018	mummified	−	−	positive
30564-2/2018	mummified	−	−	positive
37146/2018	hemorrhagic subcutaneous edema, serum hemorrhagic fluid in cavities liver degeneration	−	−	positive

**Table 2 viruses-11-00744-t002:** Comparisons of nucleotide sequences of 5′-UTR of the sheep pestiviruses identified in this study (Ovine/IT/1756/2017-CC and -EO, Ovine/IT/338710-2/2017, Ovine/IT/338710-3/2017, and Ovine/IT/16235-2/2018) and pestiviruses classified in other species. Bold lines identify strains belonging to the same species/group (nc = notclassified).

PestivirusSpecies/Strain		1-	2-	3-	4-	5-	6-	7-	8-	9-	10-	11-	12-	13-	14-	15-	16-	17-	18-	19-	20-	21-	22-	23-	24-	25-
	1. Ovine/Italy/1756/2017CC	XXX																								
	2. Ovine/Italy/1756/2017EO	100	XXX																							
	3. Ovine/Italy/16235-2/2018	99.5	99.5	XXX																						
	4. Ovine/Italy/338710-2/2017	99.5	99.5	99.1	XXX																					
	5. Ovine/Italy/338710-3/2017	94.8	94.8	95.3	94.3	XXX																				
C	6. AF091661_CSFV_strain_Brescia	87.6	87.6	88.1	87.0	91.2	XXX																			
	7. U90951_CSFV_strain_Alfort_A19	87.5	87.5	88.1	87.0	91.2	98.1	XXX																		
	8. X87939_CSFV_strain_Alfort/187	87.5	87.5	88.1	87.0	91.2	98.1	100	XXX																	
	9. FJ265020_CSFV_Sp01_Spain	88.1	88.1	88.6	87.5	90.7	95.8	95.7	95.7	XXX																
nc	10. AY159514.1_Pestivirus_strain_5440/99	95.3	95.3	95.8	94.8	98.6	90.7	90.7	90.7	90.2	XXX															
nc	11. AF462000-Tunisian_sheep_virus_35T	86.4	86.4	87.0	85.8	85.9	82.0	82.5	82.5	81.5	86.4	XXX														
G	12. AB040131.1_Pestivirus_Giraffe_H138	72.0	72.0	72.6	71.2	71.4	68.5	71.2	71.2	69.1	72.7	79.4	XXX													
I	13. KM408491_Pestivirus_Burdur/05_TR	83.6	83.6	84.2	84.2	80.8	80.1	80.0	80.0	82.4	81.3	80.1	71.2	XXX												
	14. NC_018713_Pestivirus_Aydin/04_TR	81.9	81.9	82.5	82.5	80.2	81.8	80.6	80.6	82.9	82.0	81.9	76.4	92.9	XXX											
A	15. CS114709_BVDV1_NADL	71.8	71.8	71.1	71.0	71.1	71.9	71.7	71.7	69.6	70.4	69.0	63.9	74.1	73.5	XXX										
	16. AF091605_BVDV1_strain_Oregon_C24V	70.5	70.5	71.2	69.7	72.5	71.9	71.8	71.8	70.3	73.1	69.0	63.8	72.9	71.0	93.3	XXX									
B	17. L32886_BVDV2_strain_890	70.3	70.3	70.9	69.5	66.1	64.5	65.7	65.7	65.6	66.1	68.5	68.0	65.9	62.5	72.4	69.8	XXX								
	18. AY379547_BVDV2_strain_Giessen_6	71.6	71.6	72.3	70.8	68.9	66.6	67.8	67.8	67.6	68.9	68.5	70.0	71.3	69.3	74.9	72.4	90.9	XXX							
D	19. AB122086-BDV_strain_Rudolph	79.0	79.0	79.6	78.3	79.6	82.0	81.9	81.9	81.9	79.6	78.7	70.1	83.0	81.3	67.7	65.6	61.8	64.0	XXX						
	20. U65022_BDV_strain_Moredun_cp	82.5	82.5	82.0	82.5	80.7	79.6	79.5	79.5	80.8	80.7	78.7	65.7	82.6	80.9	66.6	63.7	60.7	61.4	89.6	XXX					
	21. EF693989_BDV_isolate_90-F-6227	77.7	77.7	77.1	77.6	75.8	80.9	80.8	80.8	81.5	75.8	74.5	68.7	81.0	79.8	68.9	66.1	65.5	63.9	88.1	91.7	XXX				
	22. AY738080_BDV_Chamois	79.5	79.5	78.9	79.4	77.1	78.4	78.3	78.3	77.1	77.7	75.0	68.7	77.4	78.0	66.2	68.3	63.9	63.9	82.3	84.7	84.7	XXX			
	23. EF693998_BDV_isolate_96-F-7624	79.5	79.5	78.9	79.4	78.3	76.5	76.4	76.4	78.3	77.7	74.3	65.6	77.0	73.9	68.3	67.0	63.3	61.7	86.8	87.9	87.4	88.5	XXX		
	24. EF693996_BDV_isolate_94-F-7446/1	80.9	80.9	80.3	80.8	81.5	80.9	80.8	80.8	80.9	80.3	80.1	70.0	78.0	78.6	70.3	68.9	65.3	63.9	89.2	91.8	90.3	89.6	90.1	XXX	
	25. KT072634_BDV_isolate_Italy-103761	78.2	78.2	78.8	77.4	77.5	80.0	79.9	79.9	81.8	77.5	74.1	66.3	78.9	76.5	68.9	68.9	64.4	63.6	89.6	88.0	86.4	84.0	88.5	88.1	XXX

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
