# Peer review of "Isolation and Full-Length Sequence Analysis of a Pestivirus from Aborted Lamb Fetuses in Italy"

_viruses, 2019, doi:10.3390/v11080744_

Round 1

Reviewer 1 Report

The manuscript presents interesting data on ovine pestiviruses with close phylogenetic link to classical swine fever virus and not border disease virus. Not only 5'UTR was tested but also complete genomes of 3 viral strains were analysed. I have no remarks to study design, methods used and results obtained. Some small remarks mostly related to English language.

It is suggested to describe in abstract all pathogens analysed and not only SBV and pestiviruses. It is unclear what primers were used for molecular analysis based on work of Letellier et al. since they obtained a product of 287 bp and in the manuscript authors mention 264 bp size (line 98). Also the names of primers are confusing (line 131) where names of primers from Letellier paper and Vilcek primers are mixed together. No reference is given for MUSCLE software (line 111).
It is suggested to change word "North" to "Northern" (line 18 and later on) and word "font" (line 242) to "source" which is more common. Some words are joined together and space is missing (like lin 141: andOvine/IT... or line 196: thanto). In line 255 work "detect" should be changed to past form.

Author Response

The manuscript presents interesting data on ovine pestiviruses with close phylogenetic link to classical swine fever virus and not border disease virus. Not only 5'UTR was tested but also complete genomes of 3 viral strains were analysed. I have no remarks to study design, methods used and results obtained. Some small remarks mostly related to English language.

It is suggested to describe in abstract all pathogens analysed and not only SBV and pestiviruses.

Reply: we added in the abstract the list of pathogens investigated (see lines 21-22)

It is unclear what primers were used for molecular analysis based on work of Letellier et al. since they obtained a product of 287 bp and in the manuscript authors mention 264 bp size (line 98).

Reply The referee’s comment is correct. Indeed, the size of the amplicon is 287 and not 264 as we initially reported. Therefore, we changed it (see lines 104 and 141).

Also the names of primers are confusing (line 131) where names of primers from Letellier paper and Vilcek primers are mixed together.

Reply. The referee’s comment is correct. We used the primers of Letellier and the amplicon obtain was again 287 bp; so we changed accordingly the text and included the reference to Letellier et al., 1999 (see line 142).

No reference is given for MUSCLE software (line 111).

Reply: MUSCLE software is part of the ViPR, so the reference is the same reported in the line before. Nevertheless, we add the reference again (see line 121)

It is suggested to change word "North" to "Northern" (line 18 and later on) and word "font" (line 242) to "source" which is more common. Some words are joined together, and space is missing (like lin 141: andOvine/IT... or line 196: thanto). In line 255 work "detect" should be changed to past form.

Reply: we changed the words and separated them according to reviewer’s suggestion

Reviewer 2 Report

Authors identified pestiviruses from aborted lambs in the north Italy, and analyzed the viruses phylogenetically based on their nucleotide sequences. They were defined as "new" or "novel" pestivirus by the authors' criteria in the paper, but they were also seemed to be a different genotype of BDV. As discussed in the text by authors, it is difficult to define newly recognized pathogen. However, it is worth reporting a new branch of phylo-tree from field cases.

5'-UTR is an abbreviation of 5'-untransrated region, so the description of "5'-UTR region" (253) is inadequate.

Author Response

Authors identified pestiviruses from aborted lambs in the north Italy, and analyzed the viruses phylogenetically based on their nucleotide sequences. They were defined as "new" or "novel" pestivirus by the authors' criteria in the paper, but they were also seemed to be a different genotype of BDV. As discussed in the text by authors, it is difficult to define newly recognized pathogen. However, it is worth reporting a new branch of phylotree from field cases.

5'-UTR is an abbreviation of 5'-untransrated region, so the description of "5'-UTR region" (253) is inadequate.

Reply. We corrected as suggested, cancelling “region” either in the line indicated by the referee and in all the other cases where “5’-UTR region” was mentioned.

Reviewer 3 Report

Sozzi et al., Isolation and full-length sequence analysis of a pestivirus from aborted lamb fetuses in Italy describes clinical examination, diagnostic assessment, and whole genome sequencing of a previously unidentified variant (perhaps species) of pestivirus from an outbreak at a single large farm in Italy. The study was well conducted, generated multiple types of data and used appropriate reference samples for phylogenetic inference analyses. The data generated herein has provided a foundation for the scientific community’s understanding of this novel pestivirus.

General comments, most of which are clarified below:

The clinical data could be more transparently presented if included in Table 1 The methods section for phylogenetic analyses and tissue processing for diagnostics should be clarified Please provide higher resolution figures. It is difficult to read taxon labels in phylogenetic trees. Were there any differences between EO and CC genomes from the same case? Based on the text, only one for each case was submitted to GenBank. If genetic distance is not sufficient to meet criteria set forth by ICTV, a larger section of discussion should be reserved for how other criteria support recognition as a novel species if those data are available (in vitro data such as presence of CPE, requirement of different cell lines, etc.).

Specific comments:

Abstract

Line 16: Remove “and classified new” – using “recently discovered” and “new” is redundant

Line 17: Remove “es” from pestiviruses – strains is plural

Line 21: virus in Schmallenberg virus should be lower case

Line 24: The isolated virus sequences were not identical. There was variation (4-66 mutations) between the 3 genomes deposited in GenBank

Intro

Lines 31-35: Official species names, genera, and families of viruses should be italicized.

Lines 39 & 40: Should “rns” should be super script as shown on the ICTV website?

Line 43: Change to “not entirely” or “not completely” host-specific

Materials and Methods:

Line 63: Is this sentence stating 2000 sheep were present at that farm? Reared suggests raised from infancy to maturity. Consider rewording for clarity "head of sheep"

Line 60: Do data exist regarding number of pregnancies, aborted pregnancies, healthy births for the time frame examined?

Line 72: Were all organs pooled and homogenized together prior to testing? Or were they tested separately? It would be interesting to note which organs tested positive for different cases if those data are available.

Line 79-80: What was the length of time per passage of virus? How long post-inoculation were the final passages harvested?

Line 80: After reading the following sentences it is unclear if RT-PCR was conducted independently on cell lysate and then on homogenized organs or if they were pooled. At one point it states a single sample tested positive based on organs but could not be isolated, so I assume they were tested independently. Please clarify.

From the manuscript: "After three passages, cell lysates were by RT-PCR real-time assay to confirm the isolation of pestivirus. For this purpose, viral RNA was extracted from 200 μL of both homogenized organs and cell lysate by Trizol. Pan-pestivirus real-time RT-PCR was performed according to Office International Epizooties protocol [7]."

Line 92: virus in Schmallenberg virus should be lower case

Line 101: Adjusting the wording to “…were made with an Illumina TruSeq….” Would omit the need for Illumina to be cited again

Line 102: Prior to discussing sequence alignment, please provide information on assembly process (de novo assembly?), software/pipeline used, average coverage across each genome

Lines 107-114: Two separate phylogenetic analyses were conducted: one examined the 5’ untranslated region and one with whole genome or complete CDs. The methods of these two analyses were not completely clear. Was one analyzed in MEGA or were both aligned in MEGA and then analyzed in IQ-tree? The methods section mentions a maximum likelihood phylogenetic analysis, but the figure legend for Figure 1 states it was a neighbor joining tree. Please write methods for each analysis independently including number of taxa examined

Results

Lines 122-126: Did the two fetuses, which showed hairy fleece with abnormal yellow pigmentation have extrernal macroscopic lesions? If not is the abnormal pigmentation the only difference between those mentioned in section i) and section iii)?

Lines 122-126: Could this information be added to table 1? This would allow the reader to see which samples had clinical presentations and if any correlation was noted between clinical presentation and viral isolation

Table 1: Can a column stating which samples were tissue positive and which were tissue culture positive be added? Also, if organs were tested individually, information regarding which organs were positive would be informative, given this is a previously uncharacterized viral disease.

Lines 145-147: This portion contains information relevant to phylogenetic inference materials and methods. Please move some of this to the methods section. It also states both neighbor-joining and ML methods were used on whole genome analyses, but it appears NJ analyses were conducted on the 5’ UTR region and ML were conducted on the whole genome alignment. Is this the case? Additionally, the number of samples vary between datasets (whole genome & 5’ UTR).

Line 148: This line states both trees have similar topologies. They both show the novel Italian samples to group sister to the CSFV clade, but the relationship of Aydin PeVs to the BDV samples is different, and the variation in taxa included (Tunisian samples in 5’ UTR but no twhole genome dataset) makes it difficult to compare. There are Tunisian samples that contain ~3.5kb of the polyprotein gene. It may be more appropriate to say both trees recovered a sister relationship between the Italian novel pestiviruses and CSFV. This is tricky, as the Italian novel pestiviruses are not monophyletic in figure 1.

Figure 1 table legend: States 5 5’ UTR sequences were generated and used. There are 5 in the figure, but only 4 listed.   Both 1756 sequences included in the tree aren’t listed. Last sentence states bootstrap values above 60% are shown, but there are also values below 60% shown. There is no reference in the figure legend to the box highlighting AY159514, which is discussed later.

Figure 2 legend: Please adjust to state “Maximum Likelihood phylogenetic tree”

Line 171: Please adjust wording to “… small ruminants in Italy which were classified as a subtype of the BDV species…”

Line 175: Please add a space after %

Line 176: Please change 86.9 to 87.0 to match table 2 (or vice versa)

Line 182: Please add space after , and before “whose”

Table 2: This is a large table. It would make it easier for readers to see and compare values if the strain names were color coded in the same manner they were coded in the trees. Also, it might be easier to interpret if the average genetic distance of each group (CSFV, Novel Italian samples, etc.) were reported instead of strain to strain distances. This can be done by creating groups and calculating average between group distances in MEGA (a software already cited in the paper).

Table 2: The table was cutoff in the document receive by reviewers due to its size

Line 195: Add a space before the word Italian

Figure legend for figure 3: Adjust from “and the more related” to “and closely related”

Discussion

Line 209: Genomes were not identical

Line 209: Change “times” to “time points”

Line 219-220: Consider the following adjustment “…antigenic analyses determined ovine and caprine pestiviruses were not monotypic, but belong to two separate species, the Aydin-like pestiviruses (Pestivirus I) and the newly proposed species, Tunisian sheep pestiviruses.” – please italicize official species names

Line 224: Change Pestivirus genus to genus Pestivirus (italicizing Pestivirus) and change strictly to closely

Line 235: The Spanish strain appears to group sister to the Italian strains when analyzed phylogenetically (although only the 5’ UTR is available for this sample). If this is true, then this sample would either have to be grouped with the novel Italian samples for taxonomic purposes, or all samples would have to be subsumed into CSFV, as ICTV requires a species to be monophyletic. Can you please discuss this issue?

Line 253: Consider adjusting to “pestiviruses showed levels of divergence near the cutoff levels (12-13% in 5’ITR and 23% across the whole genome), but not clearly over the threshold suggested by ICTV to warrant the classification of …..” – These values were either grabbed from Table 2 or from an alignment I made with representative samples, please adjust if needed based on your multi-sequence alignment

Line 255: have any CSFV isolates other than the Spanish isolate been detected in sheep or anything other than swine? Citations to show this is unique would be beneficial to support the statement of difference in host origin. Is it possible this is simply spillover from pigs? Also, types of lesions have not been discussed in detail compared to CSFV.

Lines 257-258: Would this represent ongoing research or the next steps to generate additional data to support or refute these viruses as a new species?

Author Response

Sozzi et al., Isolation and full-length sequence analysis of a pestivirus from aborted lamb fetuses in Italy describes clinical examination, diagnostic assessment, and whole genome sequencing of a previously unidentified variant (perhaps species) of pestivirus from an outbreak at a single large farm in Italy. The study was well conducted, generated multiple types of data and used appropriate reference samples for phylogenetic inference analyses. The data generated herein has provided a foundation for the scientific community’s understanding of this novel pestivirus.

General comments, most of which are clarified below:

The clinical data could be more transparently presented if included in Table 1 The methods section for phylogenetic analyses and tissue processing for diagnostics should be clarified. Please provide higher resolution figures. It is difficult to read taxon labels in phylogenetic trees. Were there any differences between EO and CC genomes from the same case? Based on the text, only one for each case was submitted to GenBank. If genetic distance is not sufficient to meet criteria set forth by ICTV, a larger section of discussion should be reserved for how other criteria support recognition as a novel species if those data are available (in vitro data such as presence of CPE, requirement of different cell lines, etc.).

Specific comments:

Abstract

Line 16: Remove “and classified new” – using “recently discovered” and “new” is redundant

Reply: OK done

Line 17: Remove “es” from pestiviruses – strains is plural

Reply: OK done

Line 21: virus in Schmallenberg virus should be lower case

Reply: OK done

Line 24: The isolated virus sequences were not identical. There was variation (4-66 mutations) between the 3 genomes deposited in GenBank

Reply: it is true: we changed “identical” with “highly similar”

Intro

Lines 31-35: Official species names, genera, and families of viruses should be italicized.

Reply: OK done

Lines 39 & 40: Should “rns” should be super script as shown on the ICTV website?

Reply: OK done

Line 43: Change to “not entirely” or “not completely” host-specific

Reply: OK done

Materials and Methods:

Line 63: Is this sentence stating 2000 sheep were present at that farm? Reared suggests raised from infancy to maturity. Consider rewording for clarity "head of sheep"

Reply: OK done

Line 60: Do data exist regarding number of pregnancies, aborted pregnancies, healthy births for the time frame examined?

Reply: we added a rough estimation of the frequency of abortion as retrieved form the practitioner vet (see line 73).

Line 72: Were all organs pooled and homogenized together prior to testing? Or were they tested separately? It would be interesting to note which organs tested positive for different cases if those data are available.

Reply: unfortunately for a matter of costs of analyses, which were at that time in charge of the farmer, as diagnostic routine approach, we did not perform single organ analyses. So, the available data are referred to the pool of organs from each fetus. We changed the sentence accordingly (see line 74).  

Line 79-80: What was the length of time per passage of virus? How long post-inoculation were the final passages harvested?

Reply: the information required were added to the text and the sentence rephrased to be clearer (lines 88-90)

Line 80: After reading the following sentences it is unclear if RT-PCR was conducted independently on cell lysate and then on homogenized organs or if they were pooled. At one point it states a single sample tested positive based on organs but could not be isolated, so I assume they were tested independently. Please clarify.

From the manuscript: "After three passages, cell lysates were by RT-PCR real-time assay to confirm the isolation of pestivirus. For this purpose, viral RNA was extracted from 200 μL of both homogenized organs and cell lysate by Trizol. Pan-pestivirus real-time RT-PCR was performed according to Office International Epizooties protocol [7]."

Reply: we acknowledge the comment of the referee about the clarity of this sentence. So, we changed, integrated and rephrased it (lines 81-90)

Line 92: virus in Schmallenberg virus should be lower case

Reply: OK done

Line 101: Adjusting the wording to “…were made with an Illumina TruSeq….” Would omit the need for Illumina to be cited again

Reply: OK done

Line 102: Prior to discussing sequence alignment, please provide information on assembly process (de novo assembly?), software/pipeline used, average coverage across each genome

Reply: OK done. The requested information and data were included in the text (see lines 107-111)

Lines 107-114: Two separate phylogenetic analyses were conducted: one examined the 5’ untranslated region and one with whole genome or complete CDs. The methods of these two analyses were not completely clear. Was one analyzed in MEGA or were both aligned in MEGA and then analyzed in IQ-tree? The methods section mentions a maximum likelihood phylogenetic analysis, but the figure legend for Figure 1 states it was a neighbor joining tree. Please write methods for each analysis independently including number of taxa examined

Reply: OK, we changed the sentence according to the suggestion, trying to be clearer in defining the method used for phylogenetic analyses of 5’-UTR and complete sequences. We also integrated here what was erroneously included in the results section as pointed by the referee (see lines 116-121).

Results

Lines 122-126: Did the two fetuses, which showed hairy fleece with abnormal yellow pigmentation have external macroscopic lesions? If not is the abnormal pigmentation the only difference between those mentioned in section i) and section iii)?

Reply: in order to be clearer and answer the questions of the referee we better explained the lesions found by rephrasing the whole paragraph (lines 132-137).

Lines 122-126: Could this information be added to table 1? This would allow the reader to see which samples had clinical presentations and if any correlation was noted between clinical presentation and viral isolation

Reply; we followed the recommendation of the referee and, after having rewritten the text (lines 132-137), we also added the lesions data in table 1.

Table 1: Can a column stating which samples were tissue positive and which were tissue culture positive be added? Also, if organs were tested individually, information regarding which organs were positive would be informative, given this is a previously uncharacterized viral disease.

Reply: we differentiated the results for pestivirus according to the source (EO and CC). Unfortunately, as specified before we do not have results for separated organs. 

Lines 145-147: This portion contains information relevant to phylogenetic inference materials and methods. Please move some of this to the methods section. It also states both neighbor-joining and ML methods were used on whole genome analyses, but it appears NJ analyses were conducted on the 5’ UTR region and ML were conducted on the whole genome alignment. Is this the case? Additionally, the number of samples vary between datasets (whole genome & 5’ UTR).

Reply: OK, we changed as indicated, both the material and methods and results section, and indicated the number of taxa directly in the legends of the two figures.

Line 148: This line states both trees have similar topologies. They both show the novel Italian samples to group sister to the CSFV clade, but the relationship of Aydin PeVs to the BDV samples is different, and the variation in taxa included (Tunisian samples in 5’ UTR but no whole genome dataset) makes it difficult to compare. There are Tunisian samples that contain ~3.5kb of the polyprotein gene. It may be more appropriate to say both trees recovered a sister relationship between the Italian novel pestiviruses and CSFV. This is tricky, as the Italian novel pestiviruses are not monophyletic in figure 1.

Reply: OK. We agree with referee comments and thus we rewrote almost entirely the sentences (see lines 157-159 and 170-173)

Figure 1 table legend: States 5 5’ UTR sequences were generated and used. There are 5 in the figure, but only 4 listed.   Both 1756 sequences included in the tree aren’t listed.

Reply: we modified the legend of Figure 1, by including both sequences of strain Ovine/IT/1756/2017, i.e. the one obtained from original extracts (EO) and that obtained from cell culture (CC) (see lines 164-168.

Last sentence states bootstrap values above 60% are shown, but there are also values below 60% shown.

Reply: OK, by replacing the Figure 1 we also cancelled the values below 60%

There is no reference in the figure legend to the box highlighting AY159514, which is discussed later.

Reply: we have replaced the figure with one with higher resolution, so we changed it and cut the box highlighting the Spanish strain AY159514

Figure 2 legend: Please adjust to state “Maximum Likelihood phylogenetic tree”

Reply: OK done

Line 171: Please adjust wording to “… small ruminants in Italy which were classified as a subtype of the BDV species…”

Reply: OK done

Line 175: Please add a space after %

Reply: OK done

Line 176: Please change 86.9 to 87.0 to match table 2 (or vice versa)

Reply: OK done

Line 182: Please add space after, and before “whose”

Reply: OK done

Table 2: This is a large table. It would make it easier for readers to see and compare values if the strain names were color coded in the same manner they were coded in the trees. Also, it might be easier to interpret if the average genetic distance of each group (CSFV, Novel Italian samples, etc.) were reported instead of strain to strain distances. This can be done by creating groups and calculating average between group distances in MEGA (a software already cited in the paper).

Reply: as stated before we changed the figures in order to have them at a higher resolution. We also decided to prepare them without any colors’ difference in the text in order to delimitate the clusters/groups. This because some colors were too faint and the description of the isolates difficult to be read. Thus, we decided to highlight in bold the cells in the table 2 related to the different groups/clusters and included the reference to the established classification of the species belonging to the Pestivirus genus.

Table 2: The table was cutoff in the document receive by reviewers due to its size

Reply: OK, we reduced the size

Line 195: Add a space before the word Italian

Reply: OK done

Figure legend for figure 3: Adjust from “and the more related” to “and closely related”

Reply: OK done

Discussion

Line 209: Genomes were not identical

Reply: OK we changed accordingly (see previous comment line 27) “identical” with “highly similar”

Line 209: Change “times” to “time points”

Reply: OK done

Line 219-220: Consider the following adjustment “…antigenic analyses determined ovine and caprine pestiviruses were not monotypic, but belong to two separate species, the Aydin-like pestiviruses (Pestivirus I) and the newly proposed species, Tunisian sheep pestiviruses.” – please italicize official species names

Reply: OK done. We fully accepted the rephrasing proposed by the referee (see lines 238-240)

Line 224: Change Pestivirus genus to genus Pestivirus (italicizing Pestivirus) and change strictly to closely

Reply: OK done

Line 235: The Spanish strain appears to group sister to the Italian strains when analyzed phylogenetically (although only the 5’ UTR is available for this sample). If this is true, then this sample would either have to be grouped with the novel Italian samples for taxonomic purposes, or all samples would have to be subsumed into CSFV, as ICTV requires a species to be monophyletic. Can you please discuss this issue?

Reply: we believe that this item was already more or less discussed. However, to fulfill the request of the referee we added a sentence (see lines 256-258)

Line 253: Consider adjusting to “pestiviruses showed levels of divergence near the cutoff levels (12-13% in 5’ITR and 23% across the whole genome), but not clearly over the threshold suggested by ICTV to warrant the classification of …..” – These values were either grabbed from Table 2 or from an alignment I made with representative samples, please adjust if needed based on your multi-sequence alignment

OK done. We fully accepted the rephrasing proposed by the referee (see lines 277-279).

Line 255: have any CSFV isolates other than the Spanish isolate been detected in sheep or anything other than swine? Citations to show this is unique would be beneficial to support the statement of difference in host origin. Is it possible this is simply spillover from pigs?

Reply: to fulfill the referee requirement we modified and added a sentence in the discussion (see lines 259-268)

Also, types of lesions have not been discussed in detail compared to CSFV.

Reply: the fact that lesions observed in sheep were quite different from those observed in pigs is briefly discussed (see lines 281-288).

Lines 257-258: Would this represent ongoing research or the next steps to generate additional data to support or refute these viruses as a new species?

Reply: we better indicated what we are going to do in future studies in order to support or refute these viruses as a new species (see lines 285-288).